# EXPLORING SELECTIVE LAYER FINE-TUNING IN FEDERATED LEARNING

## ABSTRACT

Federated learning (FL) has emerged as a promising paradigm for fine-tuning foundation models using distributed data in a privacy-preserving manner. Under limited computational resources, clients often find it more practical to fine-tune a selected subset of layers, rather than the entire model, based on their task-specific data. In this study, we provide a thorough theoretical exploration of selective layer fine-tuning in FL, emphasizing a flexible approach that allows the clients to adjust their selected layers according to their local data and resources. We theoretically demonstrate that the layer selection strategy has a significant impact on model convergence in two critical aspects: the importance of selected layers and the heterogeneous choices across clients. Drawing from these insights, we further propose a strategic layer selection method that utilizes local gradients and regulates layer selections across clients. Extensive experiments on both image and text datasets demonstrate the effectiveness of the proposed strategy compared with several baselines, highlighting its advances in identifying critical layers that adapt to the client heterogeneity and training dynamics in FL.

## 1 INTRODUCTION

Foundation models (Bommasani et al., 2021), including BERT (Devlin et al., 2019), GPT (Radford et al., 2019; Brown et al., 2020), CLIP (Radford et al., 2021; Dosovitskiy et al., 2021), LLaMA (Touvron et al., 2023), and so on (Ramesh et al., 2021; Chowdhery et al., 2023), have attracted considerable attention due to their exceptional ability in handling complex tasks (Eloundou et al., 2023). When it comes to practical deployments of these models in specialized fields, fine-tuning with domain-specific data becomes critical. Nevertheless, the distributed nature of data across various users and organizations presents a challenge for centralized storage and training, as it may lead to severe privacy concerns and incur additional transmission costs. Such issues have positioned federated learning (FL) (McMahan et al., 2017) as a promising paradigm to fine-tune foundation models, aligning model enhancement with privacy preservation (Chen et al., 2023a; Kuang et al., 2023).

Generally, FL aims to learn a global model through a collaborative process where clients perform local training and upload the parameter updates to a central server for aggregation. Given that clients have limited resources (Bonawitz et al., 2019; Imteaj et al., 2022), such as computational power, communication bandwidth, and available memory, it becomes impractical for them to fine-tune the entire foundation model. Two kinds of solutions have recently emerged to tackle this challenge. The first solution employs *parameter-efficient fine-tuning* techniques (Houlsby et al., 2019; Gao et al., 2021; Hu et al., 2022; Li & Liang, 2021), which introduces additional modules integrated into foundation models and updates these modules with domain-specific data while keeping the parameters of the foundation model frozen. The second one is *selective model fine-tuning* (Lee et al., 2019a; Xu et al., 2021; Zhang et al., 2022a; Shen et al., 2021), which only selects an impactful subset of parameters for optimization to streamline the fine-tuning process under resource constraints.

This study focuses on selective model fine-tuning as it is particularly well-suited to address the inherent heterogeneity in FL, i.e., the data heterogeneity and device heterogeneity (Yang et al., 2021; Chai et al., 2019; Li et al., 2022). Specifically, clients involved in FL have non-independent and identically distributed (non-IID) data and different system resources, leading to the need to customize fine-tuning strategies to such discrepancies. For example, clients with limited computation resources may opt to update only a fraction of the model, while those with sufficient resources and

high-quality data prefer fine-tuning a large portion of the model to enhance performance. Selective model fine-tuning enables clients to adjust the chosen part of the model to be updated based on their own capabilities, providing a flexible and advanced solution to mitigating sub-optimal issues induced by the heterogeneity in FL.

The exploration of selective model fine-tuning within the context of FL, is still in its early stages. Previous studies (Shen et al., 2021; Xu et al., 2021; Lee et al., 2022; Dun et al., 2022) have concentrated on designing static strategies for subnetwork selection to improve model fine-tuning performance, without adequately considering heterogeneity among clients. To fulfill this gap, in this paper, we provide a comprehensive theoretical analysis on selective model fine-tuning in FL, focusing on a general scenario where clients are allowed to choose different layers for local training and vary their choices across different training epochs, called *selective layer fine-tuning*. Specifically, we formulate the optimization objective of selective layer fine-tuning in FL, and provide insights on effectively determining critical layers to achieve model convergence. Building on these insights, we further propose a novel layer selection strategy that leverages local gradients and the regulation of unified selections.

Our main contributions are summarized as follows:

- We study a practical FL setup where clients choose to fine-tune some layers of the model, with arbitrary layer selection that may vary among clients and across different training epochs. We theoretically analyze such a training scheme and investigate the impact of layer selection. The analytical results show that the selected layers affect the convergence performance with two critical aspects, namely the importance of layers and heterogeneous choices across clients.

- Building on the theoretical analysis, we formulate the optimization problem of selective layer fine-tuning considering the limited and diverse resource budgets of clients. Inspired by the solution to this optimization problem, we propose an effective strategy for selecting layers for fine-tuning that are well-suited for the local data and available resources at clients.

- We conduct experiments to compare the proposed layer selection strategy with baseline methods on both image and text datasets. Experimental results demonstrate the superior performance of the proposed strategy in achieving better model performance, highlighting that the proposed strategy can find critical layers for fine-tuning while considering the client heterogeneity and training dynamics in FL[1].

## 2 RELATED WORKS

Various approaches have been proposed to properly select a subset of parameters for fine-tuning foundation models within centralized training, including optimizing a non-structured mask matrix (Lee et al., 2019a; Xu et al., 2021; Zhang et al., 2022a; Shen et al., 2021; Zaken et al., 2022; Zhang et al., 2023; Kovaleva et al., 2019; Lee et al., 2019b) and adopting layer-wise selection strategies (Kovaleva et al., 2019; Lee et al., 2019b; 2022; Kaplun et al., 2023). For example, Lee et al. (2019a) suggest updating the parameters in a stochastic manner based on the Bernoulli distribution, while Kovaleva et al. (2019); Lee et al. (2019b) showcase that fine-tuning the top few layers achieves competitive model performance in downstream tasks. Moreover, Lee et al. (2022) propose to select layers according to their gradient statistics.

Recent studies have extended the selective fine-tuning techniques to FL scenarios (Nguyen et al., 2022a; Chen et al., 2022; Hilmkil et al., 2021; Zhang et al., 2022b). Specifically, researchers (Lee et al., 2023; Dun et al., 2022) investigate layer-wise network decomposition to achieve selective model fine-tuning on clients. However, these works fail to offer methodologies for adaptive and dynamic layer selection that takes into account the heterogeneous characteristics of clients. In addition, personalized FL algorithms (Pillutla et al., 2022; Chen et al., 2023b) propose to train different subnetworks on clients towards better local models. Different from previous studies, we focus on providing an in-depth understanding of selective layer fine-tuning in FL, considering the heterogeneity from the perspective of client resources and local data distributions.

---

[1]The source codes are available at `https://anonymous.4open.science/r/fed_selected_tune/`.

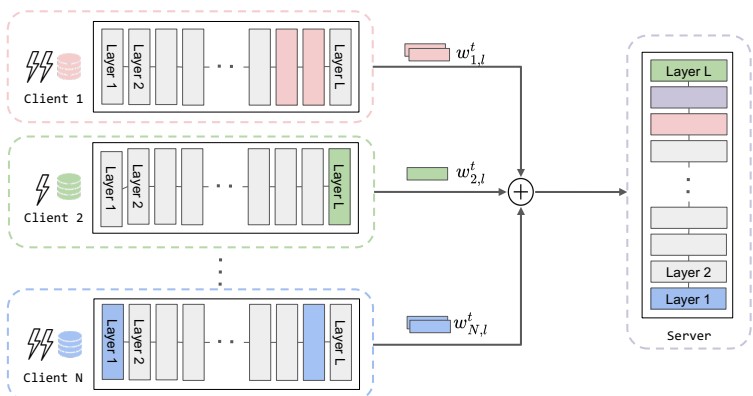

Figure 1: An overview of selective layer fine-tuning in FL. The colored layers are selected for fine-tuning.

## 3 PROBLEM FORMULATION

**Federated learning** We consider an FL system with a central server and $N$ clients (denoted by the set $\mathcal{N} = \{1, \ldots, N\}$), where each client has a private dataset $\mathcal{D}_i$ consisting of $d_i = |\mathcal{D}_i|$ data instances. The server owns a pretrained foundation model $\theta \in \mathbb{R}^P$, containing $P$ trainable parameters and $L$ layers with the index set $\mathcal{L} = \{1, 2, \ldots, L\}$. The server aims to fine-tune this foundation model based on the clients' datasets $\mathcal{D} = \{\mathcal{D}_i\}_{i \in \mathcal{N}}$ but does not directly access these datasets. The learning goal is formally given as:

$$\min_{\theta \in \mathbb{R}^P} f(\theta) = \sum_{i=1}^{N} \alpha_i f_i(\theta), \tag{1}$$

where $\alpha_i = \frac{d_i}{\sum_{j=1}^{N} d_j}$ denotes the relative sample size, and $f_i(\theta) = \frac{1}{d_i} \sum_{\xi \in \mathcal{D}_i} F_i(\theta; \xi)$ denotes the local training objective of client $i$. Here we use $F_i(\theta; \xi)$ to define the (possibly non-convex) loss function computed by the model $\theta$ on data sample $\xi$. The training process of FL is divided into $T$ training epochs. In each epoch $t \in [T]$, the server chooses a subset of clients $\mathcal{S}^t$, and sends the up-to-date global model $\theta^t$ to these clients for local training.

**Selective layer fine-tuning in FL** An overview of selective layer fine-tuning in FL is illustrated in Figure 1. Due to resource limitations, clients tend to update some of the layers in the local training process rather than the entire global model. Formally, we define a masking vector $\mathbf{m}_i^t \in \{0, 1\}^L$ for each client $i$. The $l$-th element $\mathbf{m}_i^t(l)$ equals 1 if the $l$-th layer is selected to be updated in the $t$-th training epoch, and $\mathbf{m}_i^t(l) = 0$ otherwise. Accordingly, the selected layer set of client $i$ is denoted by $\mathcal{L}_i^t \triangleq \{l \in \mathcal{L} | \mathbf{m}_i^t(l) = 1\}$, and the set for all selected layers in the $t$-th training epoch is denoted by $\mathcal{L}_t = \bigcup_{i \in \mathcal{S}^t} \mathcal{L}_i^t$. The choice of selected layer sets has a substantial effect on training performance, which will be discussed in detail later.

After determining the selected layer set $\mathcal{L}_i^t$, clients initialize the local model according to the global model sent by the server, i.e., $\theta_i^{t,0} = \theta^t$, and train the local model for $\tau$ local steps using the mini-batch SGD algorithm (McMahan et al., 2017; Wang et al., 2020; Karimireddy et al., 2020). For local step $k \in [\tau]$, client $i$ samples a batch of data instances $\xi_i^{t,k}$, and calculates the gradients for the selected layers, which is given as[2]:

$$\sum_{l \in \mathcal{L}_i^t} g_{i,l}(\theta_i^{t,k}; \xi_i^{t,k}) = \sum_{l \in \mathcal{L}_i^t} \nabla_l F_i(\theta_i^{t,k}; \xi_i^{t,k}). \tag{2}$$

Notably, the local gradient calculation pertains solely to the layers within the subset $\mathcal{L}_i^t$. Afterward, the local model is updated with the learning rate $\eta$:

$$\theta_i^{t,k} = \theta_i^{t,k-1} - \eta \sum_{l \in \mathcal{L}_i^t} g_{i,l}(\theta_i^{t,k-1}; \xi_i^{t,k-1}), \forall k \in \{1, 2, \ldots, \tau\}. \tag{3}$$

---

[2] $\nabla_l F(\theta)$ represents the gradient of a function $F(\theta)$ w.r.t. the parameters of the $l$-th layer in model $\theta$.

---

**Algorithm 1** Selective Layer Fine-tuning in FL

---

   **Input:** The pre-trained global model $\theta^0$
   **for** $t = 0, 1, \ldots, T - 1$ **do**
      Sample a set of clients $\mathcal{S}^t$;
      Broadcast the up-to-date global model $\theta^t$ and selected layer set $\mathcal{L}_i^t$ to clients $\mathcal{S}^t$;
      **for** each client $i$ in $\mathcal{S}^t$ **do**
         Compute the gradients w.r.t. layers $\mathcal{L}_i^t$ and update the model for $\tau$ steps;   {▷ Equation (3)}
         Upload the accumulated updates $\Delta_i^t$ to the server;               {▷ Equation (4)}
      **end for**
      Compute the global update $\Delta^t$;                                 {▷ Equation (5)}
      Update the global model $\theta^t$;                               {▷ Equation (6)}
   **end for**
   **Return:** The global model $\theta^T$

---

The accumulated model update in local training is summarized as:

$$\Delta_i^t = \frac{1}{\eta}(\theta_i^{t,0} - \theta_i^{t,\tau}) = \sum_{k=0}^{\tau-1} \sum_{l \in \mathcal{L}_i^t} g_{i,l}(\theta_i^{t,k}; \xi_i^{t,k}). \tag{4}$$

After local training, clients upload their model updates $\Delta_i^t, i \in \mathcal{S}^t$ to the server. The server performs federated aggregation among these model updates and optimizes the global model accordingly:

$$\Delta^t = \sum_{l \in \mathcal{L}_t} \sum_{i \in \mathcal{S}^t} w_{i,l}^t \sum_{k=0}^{\tau-1} g_{i,l}(\theta_i^{t,k}; \xi_i^{t,k}), \tag{5}$$

and

$$\theta^{t+1} = \theta^t - \eta \Delta^t. \tag{6}$$

Inspired by previous studies (McMahan et al., 2017; Li et al., 2020), the aggregation weights in selective layer fine-tuning are defined based on the data ratio and the masking vectors, which are formally given as:

$$w_{i,l}^t = \begin{cases} \frac{d_i}{\sum_{\{j \in \mathcal{S}^t \mid \mathbf{m}_j^t(l)=1\}} d_j}, & \text{if } \mathbf{m}_i^t(l) = 1, \\ 0, & \text{otherwise.} \end{cases} \tag{7}$$

The details of the training process are summarized in Algorithm 1.

## 4   Which Layers Should Be Selected For Fine-Tuning?

The aforementioned training process provides substantial flexibility in selective layer fine-tuning, namely, clients are allowed to select different layers for local training and adjust their choices in different training epochs. Such flexibility enables clients to tailor their local training to their data and resources, providing feasible solutions for handling the heterogeneity in FL.

However, without a well-designed strategy for layer selection, the optimization of the global model in FL could be severely hindered, potentially leading to a suboptimal solution or even failure in convergence. As a result, researchers have proposed several useful strategies for layer selection in recent years, including:

- All clients select the same layer set for fine-tuning (Pillutla et al., 2022; Lee et al., 2019a; Zhang et al., 2022a; 2023; Lee et al., 2019b), i.e., $\mathcal{L}_i^t = \mathcal{L}_j^t, \forall i \neq j$;
- Clients fix their selections across different training epochs (Arivazhagan et al., 2019; Chen et al., 2023b), i.e., $\mathcal{L}_i^{t_1} = \mathcal{L}_i^{t_2}, \forall t_1, t_2 \in [T]$.

These strategies for layer selection are proposed based on the insights drawn from experts' experience, serving as special instantiations of the selected layer sets $\mathcal{L}_i^t$. It is worth noting that these experience-driven strategies might not consistently yield optimal results in various FL applications, particularly considering client heterogeneity. This leads to an essential question: *How to effectively determine the task-specified layer selection strategy among a large search space of possible options?* In the rest of this section, we provide a theoretical analysis to answer this question.

## 4.1 Theoretical Analysis

Following previous theoretical analysis in FL (Wang et al., 2020; Karimireddy et al., 2020; Li et al., 2020), we begin with some necessary assumptions.

**Assumption 4.1.** *($\gamma$-Smoothness) There exists a constant $\gamma > 0$ such that for any $\theta, \theta' \in \mathbb{R}^P$, $\|\nabla f_i(\theta) - \nabla f_i(\theta')\|_2 \leq \gamma \|\theta - \theta'\|_2, \forall i \in \mathcal{N}$.*

For analyzing the effect of each layer on the model convergence, we give several assumptions for the gradient with respect to each layer $l$.

**Assumption 4.2.** *(Unbiased and variance-bounded stochastic gradient) The stochastic gradient $g_{i,l}(\theta^t; \xi_i^t)$ on a randomly sampled batch of data $\xi_i^t$ is an unbiased estimate of the full-batch gradient, i.e., $\mathbb{E}[g_{i,l}(\theta^t; \xi_i^t)] = \nabla_l f_i(\theta^t)$. Besides, there exist constants $\sigma_l > 0, \forall l \in \mathcal{L}$ such that $\|g_{i,l}(\theta^t; \xi_i^t) - \nabla_l f_i(\theta^t)\|^2 \leq \sigma_l^2, \forall i \in \mathcal{N}$ and $\sum_{l \in \mathcal{L}_t} \sigma_l^2 \leq \sigma^2$.*

The non-IID data owned by clients causes diverse gradients. In the following assumption, we state the diversity of each layer's gradient.

**Assumption 4.3.** *(Gradient diversity) There exist constants $\kappa_l > 0, \forall l \in \mathcal{L}$ such that $\|\nabla_l f(\theta^t) - \nabla_l f_i(\theta^t)\|^2 \leq \kappa_l^2, \forall i \in \mathcal{N}$.*

Here we first consider a case where $\tau = 1$ to simplify the analysis without affecting the insights on layer selection. The detailed analysis for the generalized case, i.e., $\tau > 1$, is provided in Appendix A.3.

Compared with the theoretical analysis for the standard FL settings (Wang et al., 2021; Li et al., 2020), there exist three additional challenges in selective layer fine-tuning. Firstly, since each client only updates some layers during the local training process, the aggregated gradient is no longer an unbiased estimate of the local gradient $\nabla f_i(\theta^t)$, i.e.,

$$\mathbb{E}[\Delta_i^t] = \sum_{l \in \mathcal{L}_i^t} \nabla_l f_i(\theta^t) \neq \nabla f_i(\theta^t), \tag{8}$$

where the inequality holds unless all layers are selected for fine-tuning, i.e., $\mathcal{L}_i^t = \mathcal{L}$. Secondly, since a certain layer may not be selected by all the clients, the aggregated gradient of this layer is not equivalent to the gradient computed based on the global loss function ($\sum_{l \in \mathcal{L}_t} \nabla_l f(\theta^t)$), which is given as:

$$\mathbb{E}[\Delta^t] = \sum_{l \in \mathcal{L}_t} \sum_{i \in \mathcal{S}^t} w_{i,l}^t \nabla_l f_i(\theta^t) \neq \sum_{l \in \mathcal{L}_t} \nabla_l f(\theta^t), \tag{9}$$

where the inequality holds unless all clients select the same subset of layers. Last but not least, the aforementioned gaps vary across different training epochs, making it rather complicated in the theoretical analysis.

To link the aggregated and desired gradients, we define a surrogate objective function representing the underlying loss function optimized by the clients, which is given as:

$$h_l^t(\theta) \triangleq \sum_{i \in \mathcal{S}^t} w_{i,l}^t f_i(\theta). \tag{10}$$

In essence, the layer-wise gradient of this objective function represents the update of the aggregated global update $\Delta^t$. This relationship is elaborated in the following lemma.

**Lemma 4.4.** *With Assumption 4.2, we have:*

$$\mathbb{E}[\Delta^t] = \sum_{l \in \mathcal{L}_t} \nabla_l h_l^t(\theta^t), \tag{11}$$

*where the expectation is with respect to mini-batch data sampling.*

*Proof.* We rewrite both sides of Equation (11) by using the definitions in Equations (5) and (10), and apply Assumption 4.2 to obtain the result. $\square$

As aforementioned, the underlying loss function $h_l^t(\theta)$ deviates from the desired global loss function $f(\theta)$, which hinders the optimization of the global model and may lead to suboptimal model performance. Such deviation can be quantified by the difference between the underlying update $\sum_{l\in\mathcal{L}_t}\nabla_l h_l^t(\theta^t)$ and the global gradient $\nabla f(\theta^t)$, i.e.,

$$\mathcal{E}_t \triangleq \left\| \nabla f(\theta^t) - \sum_{l\in\mathcal{L}_t} \nabla_l h_l^t(\theta^t) \right\|^2. \tag{12}$$

The term $\mathcal{E}_t$ can be further decomposed using the Jensen's inequality into two parts:

$$\mathcal{E}_t \leq 2 \left\| \nabla f(\theta^t) - \sum_{l\in\mathcal{L}_t} \nabla_l f(\theta^t) \right\|_2^2 + 2 \left\| \sum_{l\in\mathcal{L}_t} \nabla_l f(\theta^t) - \nabla_l h_l^t(\theta^t) \right\|_2^2. \tag{13}$$

*Remark* 4.5. These two terms in the right-hand side (RHS) of (13) can be interpreted as follows: (i) The first term is the difference between the gradient w.r.t. all layers and the gradient w.r.t. the selected layers. The value of this term becomes smaller when the selected layers have *larger gradients*; (ii) The second term represents the mismatch between the desired gradient computed by all clients (i.e., $\nabla_l f(\theta^t) = \sum_{i\in\mathcal{N}} \alpha_i \nabla_l f_i(\theta^t)$) and the underlying update computed by partial clients (i.e., $\nabla_l h_l^t(\theta^t) = \sum_{i\in\mathcal{S}_t} w_{i,l}^t \nabla_l f_i(\theta^t)$), resulting from *different layer choices* among clients. If some layer is selected by all clients, its corresponding term in this term can be diminished.

For a better understanding, the following lemma shows an upper bound for the value of $\mathcal{E}_t$.

**Lemma 4.6.** *With Assumption 4.3, we have:*

$$\mathcal{E}_t \leq 2 \underbrace{\left[ \left\| \sum_{l\notin\mathcal{L}_t} \nabla f(\theta^t) \right\|^2 \right]}_{\mathcal{E}_{t,1}} + 2 \underbrace{\sum_{l\in\mathcal{L}_t} \chi_{\mathbf{w}_{t,l}\|\alpha} \kappa_l^2}_{\mathcal{E}_{t,2}}, \tag{14}$$

*where* $\chi_{\mathbf{w}_{t,l}\|\alpha} \triangleq \sum_{i\in\mathcal{N}} \frac{(w_{i,l}^t - \alpha_i)^2}{\alpha_i}$.

The proofs can be found in Appendix A.1.

Next we aim to analyze the impact of layer selection on the convergence of the global model. Following previous studies (Bottou et al., 2018; Wang et al., 2020), we consider an algorithm to have achieved convergence if it converges to a stationary point of the global loss function, namely, if its expected squared gradient norm $\min_{t\in[T]} \mathbb{E}\left[ \|\nabla f(\theta^t)\|_2^2 \right]$ is zero. The following theorem and corollary show the convergence of the proposed selective layer fine-tuning framework for FL.

**Theorem 4.7.** *Define a constant* $C \triangleq 1 - 4\eta L > 0$. *With Assumptions 4.1-4.3, we have:*

$$\min_{t\in[T]} \mathbb{E}\left[ \|\nabla f(\theta^t)\|_2^2 \right] \leq \frac{2}{\eta C T}\left[ f(\theta^0) - f(\theta^*) \right] + \frac{2\gamma\eta}{C}\sigma^2 + \frac{1}{T}\sum_{t=1}^{T}\left( \frac{1}{\gamma\eta C} + 2 \right)(\mathcal{E}_{t,1} + \mathcal{E}_{t,2}), \tag{15}$$

*where* $\theta^* = \arg\min_{\theta\in\mathbb{R}^P} f(\theta)$ *is the best model with the minimal loss.*

The proofs can be found in Appendix A.2.

**Corollary 4.8.** *With the commonly selected learning rate* $\eta = \mathcal{O}\left( \frac{1}{\sqrt{T}} \right)$, *the RHS of (15) except the last term becomes zero as* $T \to \infty$. *Therefore, FL with selective layer fine-tuning may only oscillate around a stationary point of the global loss function with a non-zero error floor* $\mathcal{O}(\mathcal{E}_{t,1} + \mathcal{E}_{t,2})$.

According to Theorem 4.7 and Corollary 4.8, the training performance of the global model is degraded by the increase of $\mathcal{E}_{t,1} + \mathcal{E}_{t,2}$, in the following aspects:

- The term $\mathcal{E}_{t,1}$ indicates that it might lead to a suboptimal global model if layers with large gradient norms were not selected for fine-tuning.
- The term $\mathcal{E}_{t,2}$ shows that the consistent selections among clients promote the convergence of the global model. Specifically, if the $l$-th layer has large gradient diversity $\kappa_l$, implying significant objective bias among clients, reducing the weight divergence $\chi_{\mathbf{w}_{t,l}\|\alpha}$ helps to alleviate this term.

These findings highlight that the layer selection strategy that minimizes both terms can achieve better convergence of the global model. However, minimizing these two terms simultaneously may lead to contradictory selection decisions. Moreover, the optimal solution for minimizing the sum of $\mathcal{E}_{t,1} + \mathcal{E}_{t,2}$ is inaccessible, since the ground-truth values are intractable in practice. To resolve these challenges, in the next subsection, we propose a strategy to adaptively select layers for clients and promote the learning performance of the global model.

## 4.2 LAYER SELECTION STRATEGY

Based on the above analysis, we need to determine the selected layer sets $\{\mathcal{L}_t^i\}$ that minimize the values of $\mathcal{E}_{t,1}$ and $\mathcal{E}_{t,2}$. As both terms are hard to compute directly, we first design an approach to estimate their values.

To minimize $\mathcal{E}_{t,1}$, we prefer selecting layers with larger gradients, which can be achieved by maximizing the value of $\sum_{l \in \mathcal{L}_t} \|\nabla_l f(\theta^t)\|_2^2$. Since the norm of the global gradient $\nabla_l f(\theta^t)$ is unknown, we estimate it by using the sum of stochastic local gradients, expressed as $\sum_{i \in \mathcal{S}^t} \|g_{i,l}(\theta^t; \xi_i^t)\|_2^2$. Meanwhile, forcing the same layer selection among clients can reduce the value of $\chi_{\|\mathbf{w}_{t,l}\|_\alpha}$ and thus alleviate the term $\mathcal{E}_{t,2}$. For this purpose, we introduce the regularization term $\sum_{j \neq i} \|\mathbf{m}_i^t - \mathbf{m}_j^t\|_1$ into the optimization objective. Therefore, the selection of layers is determined by solving the following optimization problem:

$$\text{(P1)} \quad \max_{\{\mathbf{m}_i^t\}} \sum_{i \in \mathcal{S}^t} \sum_{l \in \mathcal{L}_i^t} \|g_{i,l}(\theta^t; \xi_i^t)\|_2^2 - \frac{\lambda}{2} \sum_{i \in \mathcal{S}^t} \sum_{j \neq i} \|\mathbf{m}_i^t - \mathbf{m}_j^t\|_1^2,$$

$$\text{s.t. } \mathcal{R}(\mathbf{m}_i^t) \leq R_i^t, \quad \forall i \in \mathcal{S}^t.$$

Here $\lambda \geq 0$ is a weighting constant. Specifically, a large $\lambda$ forces consistent selection across clients, while $\lambda = 0$ allows for independent choices among clients. The constraint in Problem (P1) ensures that the total cost of the selected layers meets the clients' local resource budgets $R_i^t$, and the cost function $\mathcal{R}(\cdot)$ is typically a linear function of $\mathbf{m}_i^t$.

Solving Problem (P1) further gives an effective layer selection strategy for clients. At the beginning of a training epoch, each participating client $i \in \mathcal{S}^t$ evaluates the current global model $\theta^t$ on a batch of local data and obtains the layer-wise gradient $g_{i,l}(\theta^t; \xi_i^t), \forall l \in \mathcal{L}$. Subsequently, clients upload the norms of these gradients $\|g_{i,l}(\theta^t; \xi_i^t)\|_2, \forall l \in \mathcal{L}$, which are $L$-dimensional vectors, to the server. With these values, the server can optimize the selected layer sets for clients by solving Problem (P1).

In general, the proposed layer selection strategy leads to client-specific layer sets, determined based on the estimated gradient norms. Meanwhile, a hyper-parameter $\lambda$ is used to regulate the extent to which clients are encouraged to select the same layer. In the next section, we empirically demonstrate the benefits of the proposed strategy in effectively identifying critical layers and achieving better model performance than existing methods.

## 4.3 DISCUSSIONS ON COMPUTATIONAL AND COMMUNICATION COSTS

In this section, we provide discussions on the computational and communication costs, considering a case where each client selects $R$ layers to fine-tune a model with a total of $L$ layers.

**Computational costs** Since both the proposed method and full model fine-tuning require the same forward operations, we focus on comparing the computational costs of backward operations among different methods. For simplicity, we assume each layer requires $b$ FLOPs of backward operations. The average computational costs of the proposed layer selection method are calculated as:

$$\text{Cost}_{\text{ours}} = \underbrace{b(L-1)}_{\text{Select}} + \underbrace{bR\tau}_{\text{Fine-tune}} = b(R\tau + L - 1), \tag{16}$$

where $\tau$ represents the local training steps. For comparison, fully fine-tuning a model requires the computational costs of:

$$\text{Cost}_{\text{full}} = bL\tau = \frac{L\tau}{R\tau + L - 1} \text{Cost}_{\text{ours}}. \tag{17}$$

As a result, the proposed method takes a much lower computational cost than full model fine-tuning, and the cost reduction is proportional to the number of layers and local training steps.

Meanwhile, the layer selection step in the proposed method introduces slightly additional costs of $\frac{L-R}{\tau L}\text{Cost}_{\text{full}}$, which can be further reduced by evaluating the model on a smaller volume of data or making the selection decision at a lower frequency.

**Communication costs**  The communication costs are determined by the transmitted bits during the training process. The proposed method only needs to transmit the selected layers, whose communication costs are much lower than those of full model fine-tuning that needs to upload the entire model. For example, assuming that different layers have the same number of parameters, the communication cost of the proposed method is $\frac{R}{L}$ of full model fine-tuning.

To summarize, the computational and communication costs of the proposed method are much lower than those of full model fine-tuning. More empirical evidence can be found in Section 5.3.

## 5 EXPERIMENTS

### 5.1 SETTINGS

**Datasets & Models**  We conduct a series of experiments on several widely-used image classification datasets, including CIFAR-10 (Krizhevsky & Hinton, 2009) and DomainNet (Peng et al., 2019), text classification dataset, i.e., XGLUE-NC (Liang et al., 2020), and five benchmark question-answering (QA) datasets, including SCIQ (Welbl et al., 2017), OpenbookQA (Mihaylov et al., 2018), PIQA (Bisk et al., 2020), ARC-Easy and ARC-Challenge (Bhakthavatsalam et al., 2021) datasets. More details of the adopted datasets can be found in in Appendix B.1.

As for the splitting of datasets, inspired by previous studies (Zhu et al., 2021; Kim et al., 2023), we consider two commonly observed data heterogeneity among clients: (i) *Label skew*: adopting Dirichlet distribution to allocate data samples of the CIFAR-10 dataset; (ii) *Feature skew*: adopting naturally domain shift on the DomainNet, XGLUE-NC, and QA datasets. Specifically, each QA dataset is equally divided into two subsets, with each client possessing one subset of samples from one of the five datasets. We adopt the CLIP model (Radford et al., 2021) for image classification tasks and the multi-lingual XLM-Roberta-Base (Conneau et al., 2019) model on the XGLUE-NC dataset. In addition, we train a LLaMA-2-7B (Touvron et al., 2023) model on the QA dataset.

**Server & Clients**  In the experiments, we set up an FL system with a central server and $N = 100$ clients. In each training epoch, the server randomly selects a subset of 20 clients, and broadcasts the up-to-date model to these clients for local training. Besides, there are $N = 10$ clients in the QA task and five clients are randomly chosen for training in each epoch. The resource budgets of clients are limited, which are quantified as the maximum number of layers they can fine-tune in the local training. For example, we use $R_i = 1$ to indicate that the resource of client $i$ cannot afford fine-tuning more than 1 layer. The resource budgets can be identical or heterogeneous among clients.

**Baselines**  We compare the proposed layer selection strategy with several competitive baselines, including: (i) **Top** (Kovaleva et al., 2019; Lee et al., 2019b): Clients only fine-tune the top few layers (near the output) based on their task-specific data; (ii) **Bottom** (Lee et al., 2022): Clients only fine-tune the bottom few layers (near the input) based on their task-specific data, which can be beneficial for the tasks with input-shift; (iii) **Both** (Xiao et al., 2023): Clients fine-tune an equal proportion of both the top and bottom layers, which shows the effectiveness for large language models; (iv) **SNR** (Mahsereci et al., 2017): Clients fine-tune the layers with higher signal-to-noise ratio (SNR) values, defined as the ratio of the mean of gradient elements to their variance; (v) **RGN** (Cheng et al., 2023; Lee et al., 2022): Clients fine-tune the layers with higher relative gradient norm (RGN) values, defined as the ratio of gradient norm to the parameter norm; (vi) Moreover, we consider **Full** model fine-tuning, i.e., training the entire model, as the performance benchmark. More implementation details can be found in Appendix B.2.

### 5.2 COMPARISONS

We conduct experiments with the *identical resource* scenario and the *heterogeneous resource* scenario.

**Identical resource scenario**  We first consider clients with identical computational resources, i.e., clients select the same number of layers ($R_i = R, \forall i \in \mathcal{N}$) for fine-tuning. The experimental results are shown in Table 1. From the model performance (accuracy) on the CIFAR-10 and DomainNet

Table 1: Test accuracy (%) on both image and text datasets, where each client selects $R$ layers for fine-tuning. The best results are highlighted in **bold**.

| | CIFAR-10 | | DomainNet | | XGLUE-NC | | QA | |
|---|---|---|---|---|---|---|---|---|
| | $R = 1$ | $R = 2$ | $R = 1$ | $R = 2$ | $R = 1$ | $R = 2$ | $R = 1$ | $R = 2$ |
| Full | 95.43 | | 90.27 | | 82.11 | | 65.98 | |
| Top | 93.09 | 93.61 | 87.86 | 88.32 | 69.86 | 77.05 | 63.90 | 64.44 |
| Bottom | 27.38 | 32.81 | 13.80 | 18.63 | 40.43 | 40.60 | 64.18 | 64.60 |
| Both | - | 94.96 | - | 85.48 | - | 74.65 | - | 64.41 |
| SNR | 94.47 | 90.49 | 86.38 | 87.67 | 69.11 | 79.92 | 63.80 | 64.58 |
| RGN | 92.69 | 89.48 | 88.80 | 87.19 | 74.06 | 79.48 | 63.73 | 64.70 |
| Ours | **95.47** | **96.05** | **89.37** | **89.64** | **74.95** | **80.39** | **64.71** | **65.03** |

Table 2: Test accuracy (%) on both image and text datasets, where clients have different resources ($R_i \in [1, 4]$). The best results are highlighted in **bold**.

| | CIFAR-10 | DomainNet | XGLUE-NC | QA |
|---|---|---|---|---|
| Full | 95.43 | 90.27 | 82.11 | 65.98 |
| Top | 91.22 | 89.29 | 78.17 | 64.10 |
| Bottom | 27.38 | 23.10 | 50.92 | 64.51 |
| Both | 89.91 | 86.27 | 73.01 | 64.64 |
| SNR | 75.72 | 87.34 | 78.24 | 64.51 |
| RGN | 93.83 | 88.19 | 79.36 | 64.56 |
| Ours | **95.57** | **89.39** | **80.18** | **65.80** |

datasets, we observe that the proposed strategy demonstrates notable superiority over partial layer fine-tuning baselines. Specifically, fine-tuning only one layer of the CLIP model achieves comparable performance with tuning the entire model, since the CLIP model is sufficiently powerful to extract useful features and thus requires less training on task-specific data. This also reveals that selective layer fine-tuning well meets the performance requirement within the resources of clients.

On text datasets, including XGLUE-NC and QA, the proposed layer selection strategy and RGN demonstrate similar performance, both surpassing other baseline methods (especially Top and Both) by noticeable margins. One potential explanation for this phenomenon could be that they result in similar layer selections, indicating that updating layers with higher relative gradient norms is more beneficial than other strategies, which is consistent with previous study (Lee et al., 2022). Moving a forward step, the proposed method adopts a flexible and dynamic layer selection strategy instead of fixed strategies, which leads to competitive performance.

**Heterogeneous resource scenario** Further, we conduct experiments with heterogeneous clients, i.e., clients have different local resources and thus tend to select different numbers of layers for fine-tuning. Such a heterogeneous resource scenario is more practical (Yang et al., 2021; Chai et al., 2019) and brings additional challenges for selective layer fine-tuning. Inspired by previous studies (Wang et al., 2020; Nguyen et al., 2022b), the number of layers to be fine-tuned, denoted as $R_i$ for client $i$, is sampled from a truncated half Normal distribution within $[1, 4]$.

The experimental results are shown in Table 2, from which we observe that the proposed strategy consistently shows superiority over all the baseline methods on all the datasets. Compared with baselines, the proposed strategy allows clients to flexibly determine the proper number of layers to be tuned and effectively find the most important layers. This advantage arises from enabling clients with sufficient resources to prioritize the selection of more critical layers instead of being restricted to layers in fixed positions. Overall, these experimental results demonstrate the advantage of the proposed strategy when handling heterogeneity in real-world FL applications.

### 5.3 FURTHER DISCUSSIONS

**Visualization of selected layers** For a better understanding on the proposed layer selection strategy,

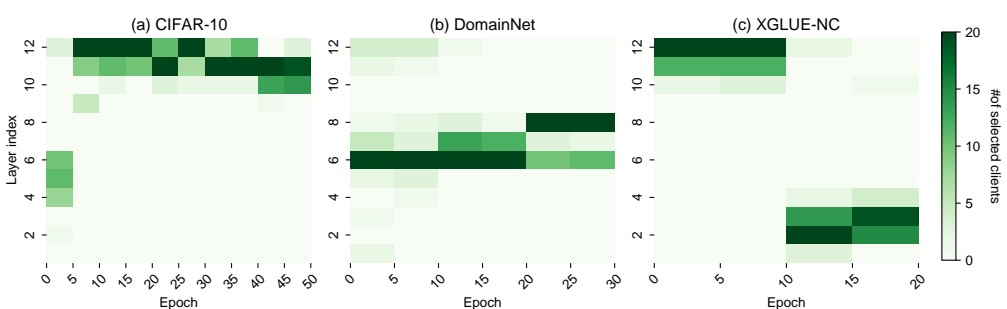

Figure 2: Visualization of selected layers ($R_i \in [1, 4]$).

Table 3: Comparisons of computational and communication costs when fine-tuning the CLIP model on the CIFAR-10 ($R = 1$). The numbers in brackets represent the costs of the proposed layer selection strategy.

|  | Computational cost (TFLOPs) | Ratio | Transmission (MBits) | Ratio |
|---|---|---|---|---|
| Full Model Fine-tuning | 8.47 | 100% | 2,811 | 100% |
| Proposed Method | 2.24 (1.51) | 26% (17%) | 234 | 8.33% |
| Proposed (Sel. Period=2) | 1.46 (0.75) | 17% (9.5%) | 234 | 8.33% |
| Proposed (Sel. Batch=1) | 0.99 (0.30) | 12% (3.4%) | 234 | 8.33% |

we visualize the selected layers on different datasets in Figure 2. When fine-tuning the CLIP model on the CIFAR-10 dataset, it can be observed that the focus is primarily on updating a few top layers, indicating that the low-level features (related to middle and bottom layers) are transferable from pre-trained data to downstream tasks. In comparison, the DomainNet dataset, characterized by a significant domain shift, necessitates extensive tuning of the middle layers in the CLIP model. Furthermore, on the XGLUE-NC dataset, we observe a clear progression of selected layers for fine-tuning, with a shift from the top layers progressively down to the bottom layers. Such a pattern is markedly different from the trend observed in the image datasets. One possible reason lies in the intrinsic differences between the modalities of text and image data. These results highlight the necessity for adaptive layer selection and adjustment strategies in FL to accommodate varying dataset properties and domain shifts.

**Comparisons regarding computational and communication costs** We compare the computational costs (in TFLOPs) and communication costs (in transmitted MBits) of the proposed method with full model fine-tuning when adopting the CLIP model on the CIFAR-10. For the proposed method, we consider fine-tuning only one layer, as it is sufficient to achieve comparable accuracy with full model fine-tuning according to Table 1. The results in Table 3 evidence a substantial decrease in both computational and communication requirements when utilizing the proposed method. Besides, we can observe that the layer selection strategy takes as low as 3.4% of the computational costs. These experimental results demonstrate that the proposed method significantly reduces both the computational and communication costs compared to full model fine-tuning.

## 6 CONCLUSIONS

In this paper, we study a practical FL setting for fine-tuning foundation models, where clients are allowed to optimize a subset of layers using their task-specific data. We carefully consider the impact of both data heterogeneity and device heterogeneity across clients, providing a comprehensive theoretical analysis of the optimization objective of selective layer fine-tuning and global model convergence. The theoretical analysis offers insights into how the selected layers influence global model training and highlights the role of layer importance and client heterogeneity. We further propose a novel strategy for layer selection that considers the local data and available resources at clients. The experimental results demonstrate that the proposed strategy outperforms baseline strategies in improving the global model training performance and even matches full model fine-tuning performance in some scenarios, showing the potential for more efficient and tailored real-world FL applications of the proposed layer selection strategy.

**Reproducibility statement** The assumptions and proofs of theoretical results in this work are given in Section 4.1 and Appendix A.1. The experimental settings are described in Section 5.1 and Appendix B. The source codes are available at `https://anonymous.4open.science/r/fed_selected_tune/`.

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

## A CONVERGENCE ANALYSIS: FULL PROOFS

### A.1 USEFUL LEMMAS

#### A.1.1 ONE-ROUND LOSS DECAY

**Lemma A.1.** *With Assumption 4.1, we have:*

$$\mathbb{E}[f(\theta^{t+1})] - \mathbb{E}[f(\theta^t)] \le \frac{1}{2\gamma}\mathcal{E}_t + \underbrace{\mathbb{E}\left\langle \sum_{l\in\mathcal{L}_t} \nabla_l h_l^t(\theta^t), \theta^{t+1} - \theta^t \right\rangle}_{\mathcal{T}_1} + \gamma \underbrace{\mathbb{E}\left[\left\|\theta^{t+1} - \theta^t\right\|^2\right]}_{\mathcal{T}_2}. \quad (18)$$

*Proof.* We begin with analyzing the loss decay by using $\gamma$-smoothness in Assumption 4.1 as follows:

$$\mathbb{E}[f(\theta^{t+1})] - \mathbb{E}[f(\theta^t)] \quad (19)$$

$$\le \mathbb{E}\langle \nabla f(\theta^t), \theta^{t+1} - \theta^t \rangle + \frac{\gamma}{2}\mathbb{E}[\|\theta^{t+1} - \theta^t\|^2] \quad (20)$$

$$= \mathbb{E}\left\langle \nabla f(\theta^t) - \sum_{l\in\mathcal{L}_t}\nabla_l h_l^t(\theta^t) + \sum_{l\in\mathcal{L}_t}\nabla_l h_l^t(\theta^t), \theta^{t+1} - \theta^t \right\rangle + \frac{\gamma}{2}\mathbb{E}[\|\theta^{t+1} - \theta^t\|^2] \quad (21)$$

$$= \underbrace{\mathbb{E}\left\langle \nabla f(\theta^t) - \sum_{l\in\mathcal{L}_t}\nabla_l h_l^t(\theta^t), \theta^{t+1} - \theta^t \right\rangle}_{\mathcal{T}_0} + \underbrace{\mathbb{E}\left\langle \sum_{l\in\mathcal{L}_t}\nabla_l h_l^t(\theta^t), \theta^{t+1} - \theta^t \right\rangle}_{\mathcal{T}_1} + \frac{\gamma}{2}\underbrace{\mathbb{E}\left[\left\|\theta^{t+1} - \theta^t\right\|^2\right]}_{\mathcal{T}_2}. \quad (22)$$

By Young's inequality, we upper bound the term $\mathcal{T}_0$ as:

$$\mathcal{T}_0 = \mathbb{E}\left\langle \nabla f(\theta^t) - \sum_{l\in\mathcal{L}_t}\nabla_l h_l^t(\theta^t), \theta^{t+1} - \theta^t \right\rangle \quad (23)$$

$$\le \frac{1}{2\gamma}\underbrace{\mathbb{E}\left[\left\|\nabla f(\theta^t) - \sum_{l\in\mathcal{L}_t}\nabla_l h_l^t(\theta^t)\right\|^2\right]}_{\mathcal{E}_t} + \frac{\gamma}{2}\mathbb{E}\left[\left\|\theta^{t+1} - \theta^t\right\|^2\right] \quad (24)$$

$$= \frac{1}{2\gamma}\mathcal{E}_t + \frac{\gamma}{2}\mathcal{T}_2. \quad (25)$$

Plugging (25) back into (22) gives the result in (18).

$\square$

#### A.1.2 ANALYZING $\mathcal{E}_t$: PROOF OF LEMMA 4.6

In this subsection, we prove the result in Lemma 4.6.

We begin with decomposing the term $\mathcal{E}_t$ using the Jensen's inequality as:

$$\mathcal{E}_t \le 2\underbrace{\|\nabla f(\theta^t) - \sum_{l\in\mathcal{L}_t}\nabla_l f(\theta^t)\|_2^2}_{\tilde{\mathcal{E}}_{t,1}} + 2\underbrace{\|\sum_{l\in\mathcal{L}_t}\nabla_l f(\theta^t) - \nabla_l h_l^t(\theta^t)\|_2^2}_{\tilde{\mathcal{E}}_{t,2}}. \quad (26)$$

For the first term $\tilde{\mathcal{E}}_{1,t}$, we directly obtain:

$$\tilde{\mathcal{E}}_{1,t} = \mathbb{E}\left[\left\|\nabla f(\theta^t) - \sum_{l\in\mathcal{L}_t}\nabla_l f(\theta^t)\right\|^2\right] = \mathbb{E}\left[\left\|\sum_{l\notin\mathcal{L}_t}\nabla_l f(\theta^t)\right\|^2\right]. \quad (27)$$

Afterwards, we derive the value of $\tilde{\mathcal{E}}_{2,t}$ as follows:

$$\tilde{\mathcal{E}}_{2,t} = \mathbb{E}\left[\left\|\sum_{l\in\mathcal{L}_t}\nabla_l f(\theta^t) - \sum_{l\in\mathcal{L}_t}\nabla_l h_l^t(\theta^t)\right\|^2\right] \tag{28}$$

$$= \mathbb{E}\left[\left\|\sum_{l\in\mathcal{L}_t}\sum_{i\in\mathcal{N}}\alpha_i\nabla_l f_i(\theta^t) - \sum_{l\in\mathcal{L}_t}\sum_{i\in\mathcal{S}_t}w_{i,l}^t\nabla_l f_i(\theta^t)\right\|^2\right] \tag{29}$$

$$= \mathbb{E}\left[\left\|\sum_{l\in\mathcal{L}_t}\sum_{i\in\mathcal{N}}\alpha_i\nabla_l f_i(\theta^t) - \sum_{l\in\mathcal{L}_t}\sum_{i\in\mathcal{N}}w_{i,l}^t\nabla_l f_i(\theta^t)\right\|^2\right] \tag{30}$$

$$= \sum_{l\in\mathcal{L}_t}\mathbb{E}\left[\left\|\sum_{i\in\mathcal{N}}\frac{w_{i,l}^t - \alpha_i}{\sqrt{\alpha_i}}\sqrt{\alpha_i}\left(\nabla_l f_i(\theta^t) - \nabla_l f(\theta^t)\right)\right\|^2\right] \tag{31}$$

$$\leq \sum_{l\in\mathcal{L}_t}\left[\sum_{i\in\mathcal{N}}\frac{(w_{i,l}^t - \alpha_i)^2}{\alpha_i}\right]\left[\sum_{i\in\mathcal{N}}\alpha_i\mathbb{E}\left[\left\|\nabla_l f_i(\theta^t) - \nabla_l f(\theta^t)\right\|^2\right]\right] \tag{32}$$

$$\leq \sum_{l\in\mathcal{L}_t}\chi_{\mathbf{w}_{t,l}\|\alpha}\kappa_l^2. \tag{33}$$

where (32) follows the Cauchy–Schwarz inequality and (33) applies Assumption 4.3.

By substituting the RHS of (27) and (33) into (26), we complete the proof. $\qquad\square$

## A.2 Convergence Analysis for Single-Step Case

In this subsection, we consider $\tau = 1$ and prove Theorem 4.7.

We derive the value of $\mathcal{T}_1$ as follows:

$$\mathcal{T}_1 = \mathbb{E}\left\langle\sum_{l\in\mathcal{L}_t}\nabla_l h_l^t(\theta^t), -\eta\sum_{l\in\mathcal{L}_t}\nabla_l h_l^t(\theta^t)\right\rangle = -\eta\mathbb{E}\left[\left\|\sum_{l\in L_t}\nabla_l h_l^t(\theta^t)\right\|^2\right]. \tag{34}$$

Afterwards, we give an upper bound for the term $\mathcal{T}_2$ as follows:

$$\mathcal{T}_2 = \left[\left\|\eta\sum_{l\in\mathcal{L}_t}\sum_{i\in\mathcal{S}^t}w_{i,l}^t g_{i,l}(\theta^t; \xi_i^t)\right\|^2\right] \tag{35}$$

$$\leq \eta^2\mathbb{E}\left[\left\|\sum_{l\in L_t}\nabla_l h_l^t(\theta^t)\right\|^2\right] + \eta^2\sigma^2, \tag{36}$$

where (36) follows Assumption 4.2.

Using the result in Lemma A.1, we have:

$$\mathbb{E}[f(\theta^{t+1})] - \mathbb{E}[f(\theta^t)]$$

$$\leq \frac{1}{2\gamma}\mathcal{E}_t - \eta\mathbb{E}\left[\left\|\sum_{l\in L_t}\nabla_l h_l^t(\theta^t)\right\|^2\right] + \gamma\left[\eta^2\mathbb{E}\left[\left\|\sum_{l\in L_t}\nabla_l h_l^t(\theta^t)\right\|^2\right] + \eta^2\sigma^2\right] \tag{37}$$

$$= \frac{1}{2\gamma}\mathcal{E}_t - \eta(1-\gamma\eta)\mathbb{E}\left[\left\|\sum_{l\in L_t}\nabla_l h_l^t(\theta^t)\right\|^2\right] + \gamma\eta^2\sigma^2. \tag{38}$$

We define a constant $C \triangleq 1 - \gamma\eta > 0$ and arrange the terms in (38) as follows:

$$\mathbb{E}\left[\left\|\sum_{l\in L_t}\nabla_l h_l^t(\theta^t)\right\|^2\right] \leq \frac{1}{\eta C}\left[\mathbb{E}[f(\theta^t)] - \mathbb{E}[f(\theta^{t+1})]\right] + \frac{1}{2\gamma\eta C}\mathcal{E}_t + \frac{\gamma\eta}{C}\sigma^2. \tag{39}$$

By Jensen's inequality, we have:

$$\mathbb{E}\left[\left\|\nabla f(\theta^t)\right\|^2\right] \tag{40}$$

$$=\mathbb{E}\left[\left\|\nabla f(\theta^t) - \sum_{l \in L_t} \nabla_l h_l^t(\theta^t) + \sum_{l \in L_t} \nabla_l h_l^t(\theta^t)\right\|^2\right] \tag{41}$$

$$\leq 2\mathbb{E}\left[\left\|\nabla f(\theta^t) - \sum_{l \in L_t} \nabla_l h_l^t(\theta^t)\right\|^2\right] + 2\mathbb{E}\left[\left\|\sum_{l \in L_t} \nabla_l h_l^t(\theta^t)\right\|^2\right] \tag{42}$$

$$=2\mathcal{E}_t + 2\mathbb{E}\left[\left\|\sum_{l \in L_t} \nabla_l h_l^t(\theta^t)\right\|^2\right]. \tag{43}$$

Combining (39) and (43) gives:

$$\mathbb{E}\left[\left\|\nabla f(\theta^t)\right\|^2\right] \leq \frac{2}{\eta C}\left[\mathbb{E}[f(\theta^t)] - \mathbb{E}[f(\theta^{t+1})]\right] + \left(\frac{1}{\gamma\eta C} + 2\right)\mathcal{E}_t + \frac{2\gamma\eta}{C}\sigma^2. \tag{44}$$

We sum up both sides of (44) over $t = 0, 1, \ldots, T-1$ and divide them by $T$ to obtain the following result:

$$\frac{1}{T}\sum_{t=1}^{T}\mathbb{E}\left[\left\|\nabla f(\theta^t)\right\|^2\right]$$

$$\leq \frac{2}{\eta CT}\left[\mathbb{E}[f(\theta^0)] - \mathbb{E}[f(\theta^T)]\right] + \frac{1}{T}\sum_{t=1}^{T}\left(\frac{1}{\gamma\eta C} + 2\right)\mathcal{E}_t + \frac{2\gamma\eta}{C}\sigma^2 \tag{45}$$

$$\leq \frac{2}{\eta CT}\left[f(\theta^0) - f(\theta^*)\right] + \frac{1}{T}\sum_{t=1}^{T}\left(\frac{1}{\gamma\eta C} + 2\right)\mathcal{E}_t + \frac{2\gamma\eta}{C}\sigma^2 \tag{46}$$

$$\leq \frac{2}{\eta CT}\left[f(\theta^0) - f(\theta^*)\right] + \frac{2\gamma\eta}{C}\sigma^2 + \frac{1}{T}\sum_{t=1}^{T}\left(\frac{1}{\gamma\eta C} + 2\right)(\mathcal{E}_{t,1} + \mathcal{E}_{t,2}). \tag{47}$$

$$\square$$

### A.3 CONVERGENCE ANALYSIS FOR MULTI-STEP CASE

Consider the general case where $\tau > 1$. We characterize the convergence in the following theorem and note that the impact of $\mathcal{E}_{t,1} + \mathcal{E}_{t,2}$ is similar to that in Theorem 4.7.

**Theorem A.2.** *Let* $C' \triangleq 1 - 4\eta\tau - 8\eta^2\gamma^2\tau(\tau-1) - 32\eta^3\gamma^2\tau^2(\tau-1) > 0$ *and* $A_\tau \triangleq \eta + 2\eta^2\gamma^2\tau(\tau-1) + 8\eta^3\gamma^2\tau^2(\tau-1)$. *With Assumptions 4.1-4.3, we have:*

$$\frac{1}{T}\sum_{t=1}^{T}\mathbb{E}\left[\left\|\nabla f(\theta^t)\right\|^2\right] \leq \frac{2}{\eta\tau C'T}\left[f(\theta^0) - f(\theta^*)\right] + \frac{4A_\tau}{C'}\sigma^2 + \frac{1}{T}\sum_{t=1}^{T}\left(\frac{1}{\eta\tau\gamma C'} + 2\right)(\mathcal{E}_{t,1} + \mathcal{E}_{t,2}). \tag{48}$$

*Proof.* In Lemma A.1, the term $\mathcal{T}_1$ is related to client drift caused by multiple local SGD steps, which can be upper bounded as follows:

$$\mathcal{T}_1$$

$$= -\eta\sum_{k=0}^{\tau-1}\mathbb{E}\left\langle\sum_{l \in \mathcal{L}_t}\nabla_l h_l^t(\theta^t), \sum_{l \in \mathcal{L}_t}\sum_{i \in \mathcal{N}}w_{i,l}^t\nabla_l f_i(\theta_i^{t,k})\right\rangle \tag{49}$$

$$= -\eta\sum_{k=0}^{\tau-1}\mathbb{E}\left\langle\sum_{l \in \mathcal{L}_t}\nabla_l h_l^t(\theta^t), \sum_{l \in \mathcal{L}_t}\nabla_l h_l^t(\theta^t)\right\rangle$$

$$+ \eta \sum_{k=0}^{\tau-1} \mathbb{E} \left\langle \sum_{l \in \mathcal{L}_t} \nabla_l h_l^t(\theta^t), \sum_{l \in \mathcal{L}_t} \nabla_l h_l^t(\theta^t) - \sum_{l \in \mathcal{L}_t} \sum_{i \in \mathcal{N}} w_{i,l}^t \nabla_l f_i(\theta_i^{t,k}) \right\rangle \tag{50}$$

$$\leq - \frac{\eta}{2} \sum_{k=0}^{\tau-1} \mathbb{E} \left[ \left\| \sum_{l \in \mathcal{L}_t} \nabla_l h_l^t(\theta^t) \right\|^2 \right] + \frac{\eta}{2} \sum_{k=0}^{\tau-1} \mathbb{E} \left[ \left\| \sum_{l \in \mathcal{L}_t} \nabla_l h_l^t(\theta^t) - \sum_{l \in \mathcal{L}_t} \sum_{i \in \mathcal{N}} w_{i,l}^t \nabla_l f_i(\theta_i^{t,k}) \right\|^2 \right] \tag{51}$$

$$= - \frac{\eta \tau}{2} \mathbb{E} \left[ \left\| \sum_{l \in \mathcal{L}_t} \nabla_l h_l^t(\theta^t) \right\|^2 \right] + \frac{\eta}{2} \sum_{k=0}^{\tau-1} \mathbb{E} \left[ \left\| \sum_{l \in \mathcal{L}_t} \sum_{i \in \mathcal{N}} w_{i,l}^t \nabla_l f_i(\theta^t) - \sum_{l \in \mathcal{L}_t} \sum_{i \in \mathcal{N}} w_{i,l}^t \nabla_l f_i(\theta_i^{t,k}) \right\|^2 \right] \tag{52}$$

$$\leq - \frac{\eta \tau}{2} \mathbb{E} \left[ \left\| \sum_{l \in \mathcal{L}_t} \nabla_l h_l^t(\theta^t) \right\|^2 \right] + \frac{\eta \gamma^2}{2} \underbrace{\sum_{k=0}^{\tau-1} \mathbb{E} \left[ \left\| \sum_{l \in \mathcal{L}_t} \sum_{i \in \mathcal{N}} w_{i,l}^t \left( \theta^t - \theta_i^{t,k} \right) \right\|^2 \right]}_{\mathcal{T}_4}, \tag{53}$$

where (51) uses the inequality $\langle \mathbf{a}, \mathbf{b} \rangle \leq \frac{\|\mathbf{a}\|^2}{2} + \frac{\|\mathbf{b}\|^2}{2}$, and (53) follows Assumption 4.1.

Then we analyze the term $\mathcal{T}_2$ as follows:

$$\mathcal{T}_2$$

$$\leq \eta^2 \mathbb{E} \left[ \left\| \sum_{k=0}^{\tau-1} \sum_{l \in \mathcal{L}_t} \sum_{i \in \mathcal{N}} w_{i,l}^t \nabla_l f_i(\theta_i^{t,k}) \right\|^2 \right] + \eta^2 \tau \sigma^2 \tag{54}$$

$$\leq \eta^2 \tau \sum_{k=0}^{\tau-1} \mathbb{E} \left[ \left\| \sum_{l \in \mathcal{L}_t} \sum_{i \in \mathcal{N}} w_{i,l}^t \nabla_l f_i(\theta_i^{t,k}) - \sum_{l \in \mathcal{L}_t} \sum_{i \in \mathcal{N}} w_{i,l}^t \nabla_l f_i(\theta^t) + \sum_{l \in \mathcal{L}_t} \sum_{i \in \mathcal{N}} w_{i,l}^t \nabla_l f_i(\theta^t) \right\|^2 \right]$$
$$+ \eta^2 \tau \sigma^2 \tag{55}$$

$$\leq 2\eta^2 \tau \sum_{k=0}^{\tau-1} \mathbb{E} \left[ \left\| \sum_{l \in \mathcal{L}_t} \sum_{i \in \mathcal{N}} w_{i,l}^t \nabla_l f_i(\theta_i^{t,k}) - \sum_{l \in \mathcal{L}_t} \sum_{i \in \mathcal{N}} w_{i,l}^t \nabla_l f_i(\theta^t) \right\|^2 \right]$$
$$+ 2\eta^2 \tau \sum_{k=0}^{\tau-1} \mathbb{E} \left[ \left\| \sum_{l \in \mathcal{L}_t} \nabla_l h_l^t(\theta^t) \right\|^2 \right] + \eta^2 \tau \sigma^2 \tag{56}$$

$$\leq 2\eta^2 \gamma^2 \tau \sum_{k=0}^{\tau-1} \mathbb{E} \left[ \left\| \sum_{l \in \mathcal{L}_t} \sum_{i \in \mathcal{N}} w_{i,l}^t \left( \theta_i^{t,k} - \theta^t \right) \right\|^2 \right] + 2\eta^2 \tau^2 \mathbb{E} \left[ \left\| \sum_{l \in \mathcal{L}_t} \nabla_l h_l^t(\theta^t) \right\|^2 \right] + \eta^2 \tau \sigma^2 \tag{57}$$

$$= 2\eta^2 \gamma^2 \tau \mathcal{T}_4 + 2\eta^2 \tau^2 \mathbb{E} \left[ \left\| \sum_{l \in \mathcal{L}_t} \nabla_l h_l^t(\theta^t) \right\|^2 \right] + \eta^2 \tau \sigma^2, \tag{58}$$

where (54) follows Assumption 4.2, (55)-(56) apply the Jensen's inequality, and (57) follows Assumption 4.1.

Following Lemma 22 in (Pillutla et al., 2022), $\mathcal{T}_4$ can be upper bounded as:

$$\mathcal{T}_4 \leq \sum_{k=0}^{\tau-1} \mathbb{E} \left[ \left\| \sum_{l \in \mathcal{L}_t} \sum_{i \in \mathcal{N}} w_{i,l}^t \theta^t - \theta_i^{t,k} \right\|^2 \right] \tag{59}$$

$$\leq 8\eta^2 \tau^2 (\tau-1) \mathbb{E} \left[ \left\| \sum_{l \in \mathcal{L}_t} \sum_{i \in \mathcal{N}} w_{i,l}^t \nabla_l f_i(\theta^t) \right\|^2 \right] + \sum_{l \in \mathcal{L}_t} \sum_{i \in \mathcal{N}} w_{i,l}^t 4\eta^2 \tau^2 (\tau-1) \sigma_l^2 \tag{60}$$

$$= 8\eta^2 \tau^2 (\tau-1) \mathbb{E} \left[ \left\| \sum_{l \in \mathcal{L}_t} \nabla_l h_l^t(\theta^t) \right\|^2 \right] + 4\eta^2 \tau^2 (\tau-1) \sigma^2. \tag{61}$$

Plugging (54),(58) and (61) back into (18), we have the following result:

$$\mathbb{E}[f(\theta^{t+1})] - \mathbb{E}[f(\theta^t)] \tag{62}$$

$$\leq \frac{1}{2\gamma}\mathcal{E}_t - \frac{\eta\tau}{2}\mathbb{E}\left[\left\|\sum_{l\in\mathcal{L}_t}\nabla_l h_l^t(\theta^t)\right\|^2\right] + \frac{\eta\gamma^2}{2}\mathcal{T}_4 + 2\eta^2\gamma^2\tau\mathcal{T}_4 + 2\eta^2\tau^2\mathbb{E}\left[\left\|\sum_{l\in\mathcal{L}_t}\nabla_l h_l^t(\theta^t)\right\|^2\right] + \eta^2\tau\sigma^2 \tag{63}$$

$$= \frac{1}{2\gamma}\mathcal{E}_t - \frac{\eta\tau}{2}(1 - 4\eta\tau)\mathbb{E}\left[\left\|\sum_{l\in\mathcal{L}_t}\nabla_l h_l^t(\theta^t)\right\|^2\right] + \eta^2\tau\sigma^2 + \left(\frac{\eta\gamma^2}{2} + 2\eta^2\gamma^2\tau\right)\mathcal{T}_4 \tag{64}$$

$$= \frac{1}{2\gamma}\mathcal{E}_t - \frac{\eta\tau}{2}(1 - 4\eta\tau)\mathbb{E}\left[\left\|\sum_{l\in\mathcal{L}_t}\nabla_l h_l^t(\theta^t)\right\|^2\right] + \eta^2\tau\sigma^2$$

$$+ \left(\frac{\eta\gamma^2}{2} + 2\eta^2\gamma^2\tau\right)\left\{8\eta^2\tau^2(\tau-1)\mathbb{E}\left[\left\|\sum_{l\in\mathcal{L}_t}\nabla_l h_l^t(\theta^t)\right\|^2\right] + 4\eta^2\tau^2(\tau-1)\sigma^2\right\} \tag{65}$$

$$= \frac{1}{2\gamma}\mathcal{E}_t - \frac{\eta\tau}{2}\left[1 - 4\eta\tau - 8\eta^2\gamma^2\tau(\tau-1) - 32\eta^3\gamma^2\tau^2(\tau-1)\right]\mathbb{E}\left[\left\|\sum_{l\in\mathcal{L}_t}\nabla_l h_l^t(\theta^t)\right\|^2\right]$$

$$+ \left(\eta^2\tau + 2\eta^3\gamma^2\tau^2(\tau-1) + 8\eta^4\gamma^2\tau^3(\tau-1)\right)\sigma^2. \tag{66}$$

Let $C' \triangleq 1 - 4\eta\tau - 8\eta^2\gamma^2\tau(\tau-1) - 32\eta^3\gamma^2\tau^2(\tau-1) > 0$ and $A_\tau \triangleq \eta + 2\eta^2\gamma^2\tau(\tau-1) + 8\eta^3\gamma^2\tau^2(\tau-1)$. We have:

$$\mathbb{E}\left[\left\|\sum_{l\in\mathcal{L}_t}\nabla_l h_l^t(\theta^t)\right\|^2\right] \leq \frac{2}{\eta\tau C'}\left[\mathbb{E}[f(\theta^t)] - \mathbb{E}[f(\theta^{t+1})]\right] + \frac{1}{\eta\tau\gamma C'}\mathcal{E}_t + \frac{2}{C'}A_\tau\sigma^2. \tag{67}$$

Using the result in (43), we have:

$$\mathbb{E}\left[\|\nabla f(\theta^t)\|^2\right] \leq \frac{4}{\eta\tau C'}\left[\mathbb{E}[f(\theta^t)] - \mathbb{E}[f(\theta^{t+1})]\right] + \left(\frac{1}{\eta\tau\gamma C'} + 2\right)\mathcal{E}_t + \frac{4A_\tau}{C'}\sigma^2. \tag{68}$$

We sum up both sides of (68) over $t = 0, 1, \ldots, T-1$ and divide them by $T$ to obtain the following result:

$$\frac{1}{T}\sum_{t=1}^{T}\mathbb{E}\left[\|\nabla f(\theta^t)\|^2\right]$$

$$\leq \frac{2}{\eta\tau C'T}\left[\mathbb{E}[f(\theta^0)] - \mathbb{E}[f(\theta^T)]\right] + \frac{1}{T}\sum_{t=1}^{T}\left(\frac{1}{\eta\tau\gamma C'} + 2\right)\mathcal{E}_t + \frac{4A_\tau}{C'}\sigma^2 \tag{69}$$

$$\leq \frac{2}{\eta\tau C'T}\left[f(\theta^0) - f(\theta^*)\right] + \frac{1}{T}\sum_{t=1}^{T}\left(\frac{1}{\eta\tau\gamma C'} + 2\right)\mathcal{E}_t + \frac{4A_\tau}{C'}\sigma^2 \tag{70}$$

$$\leq \frac{2}{\eta\tau C'T}\left[f(\theta^0) - f(\theta^*)\right] + \frac{4A_\tau}{C'}\sigma^2 + \frac{1}{T}\sum_{t=1}^{T}\left(\frac{1}{\eta\tau\gamma C'} + 2\right)(\mathcal{E}_{t,1} + \mathcal{E}_{t,2}). \tag{71}$$

$\square$

## B EXPERIMENTAL DETAILS

We implement all methods with PyTorch and run experiments on Nvidia V100 GPUs. For fair comparisons, we adopt the same training epochs and hyper-parameters for all methods.

### B.1 TRAINING TASKS

To evaluate the methods in various scenarios with different non-IID patterns, we consider two image classification tasks and two text classification tasks.

Table 4: Summary of datasets.

| Dataset | Data Type | Non-IID Type | Partition |
|---------|-----------|--------------|-----------|
| CIFAR-10 | Image | Label skew | $Dir(0.1)$ |
| DomainNet | Image | Feature skew | Domain |
| XGLUE-NC | Text | Feature skew | Domain |
| QA | Text | Feature skew | Domain |

The image classification tasks include:

- **CIFAR-10** (Krizhevsky & Hinton, 2009): In this training task, we consider the label-skewed case where $P(y_i)$ is different among clients. Following previous works (Li et al., 2022), we adopt Dirichlet allocation with concentration parameter $\alpha = 0.1$ among clients.
- **DomainNet** (Peng et al., 2019): DomainNet contains six domains of data samples, including clipart, real, sketch, infograph, painting, and quickdraw. In this training task, we consider the varying feature case where $P(x_i|y_i)$ is different among clients. Following (Li et al., 2021), each client is allocated random samples from only one domain.

On both tasks, we fine-tune a CLIP Vision Transformer (CLIP) model (Radford et al., 2021).

Besides, we fine-tune an XLM-Roberta-Base model on the following text dataset:

- **XGLUE-NC** (Liang et al., 2020): This is a news classification task consisting of 10 classes. The news texts comprise five languages (English, Spanish, French, German, and Russian). We allocate one random language to each client, which naturally introduces domain shift among clients.

In addition, we evaluate a LLaMA-2-7B model on the QA datasets.

- **QA**: The QA datasets consist of four commonly used question-answering datasets, i.e., SCIQ (Welbl et al., 2017), OpenbookQA (Mihaylov et al., 2018), PIQA (Bisk et al., 2020), ARC-Easy and ARC-Challenge (Bhakthavatsalam et al., 2021). The datasets are transformed into classification tasks where the model determines the correct answer for each question and corresponding choices. We allocate the samples from one random dataset to each client, indicating the domain shift among clients.

### B.2 IMPLEMENTATION DETAILS

The CLIP model is pre-trained on the DataComp dataset and is adapted from `https://github.com/openai/CLIP`; the XLM-Roberta-Base model is adapted from `https://huggingface.co/xlm-roberta-base`; the LLaMA-2-7B model is adapted from `https://huggingface.co/meta-llama/LLaMA-7b-chat-hf`. In all training tasks, we freeze the embedding layers of the model and fix the classifier as commonly selected layers (Lee et al., 2019b). The values of adopted hyperparameters are summarized in Table 5. For the proposed method, we tune the value of $\lambda$ from $\{1, 5, 10, 100, 500, 1000\}$.

Table 5: Implementation details.

| Dataset | CIFAR-10 | DomainNet | XGLUE-NC | QA |
|---|---|---|---|---|
| Model | CLIP | CLIP | XLM-Roberta-Base | LLaMA-2-7B |
| Batch size | 64 | 64 | 32 | 16 |
| Learning rate | 0.01 | 0.01 | 0.01 | 2e-5 |
| Local steps* | 5 | 1 | -1 | -1 |
| Total epochs | 50 | 30 | 20 | 2 |
| $\lambda$ | 1000 | 10 | 10 | 5 |

*The local steps -1 means clients iterate all training samples (i.e., one single local training epoch).

