# OpenReview forum: "Exploring Selective Layer Fine-Tuning in Federated Learning"
_ICLR.cc/2025/Conference — Submitted to ICLR 2025_

### Official Review · Reviewer_6kD9 · 2024-10-18

**Soundness:** 2
**Presentation:** 3
**Contribution:** 2
**Rating:** 3
**Confidence:** 4

**Summary:**

This work proposes a layer-selection-based strategy to finetune large models in a federated setting. "Which layers to finetune" is selected based on the resource budget and data distribution of each client. Empirical evaluation shows the improvements in model accuracy, and theoretical analysis shows how heterogeneity based on data and hence derived gradients plays a role in global model convergence.

**Strengths:**

1. The use case is relevant and timely; and the motivation makes sense.

2. The work considers data and device heterogeneity under the FL setting, making it more practical.

3. The manuscript gives new insights on layer selection strategies related to picking layers with large gradient norms and consistency in layer selection by a client.

4. The writing is mostly clear (clarifying questions on a few confusing points have been asked in the questions section).

**Weaknesses:**

1) Lack of strong baselines. The work is compared against only finetuning top or bottom layers and against methods that select layers based on signal-to-noise ratio (SNR) and relative gradient norm (RGN); however, there are methods like FedBiot [1], which uses LoRA adapters to finetune LLMs in FL. (Although FedBiot was quite recently published, a similar baseline that includes parameter-efficient finetuning should be an essential method to compare against.)

[1] Fedbiot: LLM local fine-tuning in federated learning without full model (ACM SIGKDD 2024)

2) Which party (server or clients) selected which layers should be trained by which client is unclear from algorithm 1 and its corresponding discussion text. Line 42 and 199 say "clients tend to update some of the layers" and "clients are allowed to select different layers", but Line 166 in Algorithm 1 indicates that the server is sending the selected layers to each client, and Line 355 states "the server can optimize the selected layer sets." What would be the difference in performance and variables to consider if it's (a) the server or (b) the clients picking which layers to finetune? (I have follow-up questions in Q3 and Q4 under the "Questions" as well.)

3) Section 4.3 can be massively improved by drawing comparisons of the computational and communication costs of the proposed method with all its baselines instead of just picking the most intensive baseline of full model finetuning.

4.a) The insights into why the proposed method finetunes the CLIP model so well compared to other baselines do not make sense. The line 460 "since the CLIP model is sufficiently powerful to extract useful features..." is the property of the model, not of the proposed method. And if that's the reasoning, at least one of the Top/Bottom baselines should also have great performance, but that is not the case. According to Figure 2, Top baseline should have worked way better.

4.b) Lines 465-467 state that RGN and the proposed layer selection strategy result in similar layer selections, so why does RGN not have higher resultant accuracy? What are the differences between RGN and your proposed method?

**Questions:**

1) The authors mention two types of solutions for finetuning LLMs in a resource-constrained setting: (a) parameter-efficient finetuning (PEFT) and (b) Selective model finetuning. Since they are both orthogonal, I wonder if they can be combined for an even better performance. Also, what is the rationale behind not comparing the proposed method against PEFT methods like LoRA?

2.a) Under "Introduction," right above the list of "main contributions," there's a sentence stating "a novel layer selection strategy that leverages local gradients and the regulation of unified selections." Can the authors please clarify how their strategies are novel in terms of leveraging local gradients? Don't the SNR and RGN baselines also leverage the same local gradients?

2.b) Also, what does "unified selections" mean in this context? Is it the same client picked by the same client every round?

3) Layer selection should depend on the heterogeneity among selected clients for the current round. If clients are selecting which layers they want to finetune, is there any way for the clients to know which layers are getting selected by other clients?

4) Referring to Line 200-201, what is the rationale behind "clients being able to select which layers to finetune" relating to "handling heterogeneity in FL"? Can't/won't the clients just select the "easy to achieve high accuracy on" layers for local finetuning (which might be detrimental to minimizing the global loss)? In other words, clients (in contrast to server) selecting which layers to finetune should lead to clients being selfish and only caring about personalized accuracy, which is opposite to how global balance can be achieved in the face of heterogeneity.

5) May I know what does a "Layer" in the context of language models and the CLIP model consists of? Is it one linear layer (e.g., one of the query, key, or value linear layers) or a module containing multiple linear or other layers?

6) Theorem 4.7 has good findings (in Lines 317-323) related to the terms $\varepsilon_{t, 1}$ and $\varepsilon_{t, 2}$; I am wondering if there's an empirical way to observe these two phenomena:

(i) The selected layers have large gradient norms based on the gradients derived by the current round's clients.

(ii) Inconsistent selection of clients not leading to a global convergence.

7) What is the reasoning behind defining a resource budget in terms of how many layers a client can finetune? What if the budget is 1 layer, but that layer has large dimensions which cannot be fit into a device like a resource-constrained mobile phone? Who decides that budget and how?

8.a) Does Equation 16 have cost of $b$ in "Select" due to clients having to compute the gradient norm?

8.b) Line 383 states, "The proposed method only needs to transmit the selected layers." However, don't the clients also need to transmit the gradient norms to the server?

---

### Official Review · Reviewer_CSqE · 2024-10-28

**Soundness:** 3
**Presentation:** 3
**Contribution:** 1
**Rating:** 3
**Confidence:** 4

**Summary:**

This paper examines selective layer fine-tuning in federated learning (FL) to adapt foundation models efficiently across clients with varying resources. It presents a theoretical analysis, providing insights towards the proposed gradient-based layer selection strategy over heterogeneous clients. This strategy enables clients to choose layers based on data and resource constraints. Experiments validate its efficiency, demonstrating performance on par with full model fine-tuning in resource-limited and heterogeneous FL settings.

**Strengths:**

- The problem this paper focuses on is important. Leveraging pre-trained foundation models in federated learning often does require careful training strategies due to clients' resource limitations.
- Theoretical analysis provides insights towards the proposed layer selection strategy, indicating the effect of large gradient norms and consistent selection among clients.
- Experiments include different modalities on various datasets.

**Weaknesses:**

- While selective layer fine-tuning and gradient magnitude-based selection are useful, these approaches are both well studied in the field (e.g. [1,2]). This paper lacks a distinct methodological innovation that would set it apart from merely combining the two existing techniques.

- The theoretical insights presented do not appear specific to the layer-selection strategy, making them applicable to more granular selection approaches, such as element-wise weight selection [2]. Without a clear motivation for focusing on layer-based selection, the paper does not sufficiently justify the connection between the theoretical findings and the layer-wise selection method.

- The experimental gains reported, typically less than 1%, may not justify the complexity of implementing this approach in real-world applications. Additionally, the lack of baseline results from purely local training (without FL) limits the evaluation of the added value of the proposed method over simple local fine-tuning.

- There is no ablation study on $\lambda$. Since $\lambda$ plays a key role in controlling the degree of uniformity in layer selection across clients, an ablation study would have provided critical insights into its impact on model performance. Exploring how $\lambda$ values affect the trade-off between personalization and global model alignment could help validate the robustness of the approach, making the results more interpretable and the method more adaptable to diverse FL settings.

- I do not think $\sum_{l\in \mathcal{L}_i^t}$ is accurate. E.g. in Eq.(2) how do you add the gradient with respect to different layers together?

[1] Surgical fine-tuning improves adaptation to distribution shifts

[2] FedSelect: Personalized Federated Learning with Customized Selection of
Parameters for Fine-Tuning

**Questions:**

- Based on your theoretical findings, what is the specific reason on focusing on a layer-wise selection as opposed to element-wise selection?
- How does the pretrained foundation model's zero shot ability or local training performance compare to the federated methods?
- How does $\lambda$ affect the performance numerically? Since $\varepsilon_{t,1}$ and $\varepsilon_{t,2}$ can sometimes conflict with each other, is the performance bad when $\lambda$ is either too large or too small?

---

### Official Review · Reviewer_rcXp · 2024-10-29

**Soundness:** 2
**Presentation:** 3
**Contribution:** 3
**Rating:** 5
**Confidence:** 4

**Summary:**

This work proposes a layer selection strategy based on client gradient norms, where layers with larger gradient norms are fine-tuned rather than adjusting the entire model. This selective strategy considers practical factors such as data heterogeneity and client resources, achieving improved model performance on image and text datasets.

**Strengths:**

1. The idea of selectively fine-tuning specific layers in federated learning is innovative.
2. The paper sets up various Non-IID scenarios on four benchmark datasets.
3. The paper is well written and easy to follow.

**Weaknesses:**

1. The paper focuses on layer selection for fine-tuning but lacks a detailed discussion of experimental results in Tables 1 and 2. For instance, in the setup of selecting the same number of layers in Table 1, the differences in selected layers between the proposed method and RGN or SNR methods should be elaborated.

2. The comparative experiments lack stronger baselines, such as adding the same regularization terms to SNR and RGN as in the proposed method.

3. The paper lacks robust conclusions regarding layer selection. For instance, a dynamic layer selection strategy where bottom layers have larger gradient norms in early training stages and top layers have larger norms in later stages could be considered. Additionally, there is an absence of discussion on whether forcing all clients to select the same layers or allowing them to select different layers could be more effective.

**Questions:**

1. Layers with larger gradient norms are assumed to be more relevant to client tasks. However, could selecting these layers for global aggregation and fine-tuning exacerbate the discrepancy between global and local parameters?

2. Due to non-iid distributions, different clients should ideally select different layers. However, this work enforces the same layer selection across clients using regularization terms. Please explain the rationale behind optimization objective P1.

3. According to Section 3, only the selected layers are updated during global fine-tuning, while other layers remain unchanged. Could updating the unselected layers based on local gradients provide additional performance gains?

4. Could Figure 2 be supplemented with visualizations of the selected layers for the RGN and SNR baseline?

5. Regarding the proposed method’s reduction in computational costs in table 3, is this saving limited to the parameter updates on the client side? In the process of calculating local gradients, are gradients for all layers still required?

---

### Official Review · Reviewer_F7YF · 2024-10-31

**Soundness:** 2
**Presentation:** 3
**Contribution:** 2
**Rating:** 5
**Confidence:** 3

**Summary:**

This study investigates the concept of selective layer fine-tuning within the federated learning (FL) paradigm, where clients fine-tune only specific layers of foundation models based on their local data and resource constraints. By allowing clients to choose which layers to update, the paper emphasizes a flexible approach that adapts to the diverse capabilities and data distributions of clients. Theoretical analysis shows that the selection of layers significantly influences model convergence, focusing on the importance of chosen layers and the heterogeneity in client selections. An effective layer selection strategy leveraging local gradients is proposed, demonstrating enhanced performance in extensive experiments across image and text datasets compared to baseline methods.

**Strengths:**

1. The paper provides a robust theoretical exploration of selective layer fine-tuning, enhancing understanding of how layer selection impacts convergence in FL.

2. By allowing clients to customize their fine-tuning strategies based on local resources and data, the proposed approach effectively addresses the inherent heterogeneity present in FL scenarios.

**Weaknesses:**

1. The proposed layer selection strategy may require sophisticated implementations and tuning, which could complicate its deployment in practice, particularly for clients with limited expertise or resources.

2. The effectiveness of selective layer fine-tuning in larger, more complex models or across a significantly larger number of clients remains to be fully explored, raising questions about its scalability.

3. While the paper addresses client heterogeneity, it does not fully consider the impact of changing data distributions over time, which could affect the effectiveness of the fine-tuning strategy as client environments evolve.

**Questions:**

See the weaknesses.

---

### Official Review · Reviewer_6bft · 2024-11-03

**Soundness:** 2
**Presentation:** 4
**Contribution:** 3
**Rating:** 5
**Confidence:** 4

**Summary:**

This paper consider the selective layer update in federated learning. It first theoretically analyzes the impact of layer selection on the model convergence. Then, motivated by the form of the error term, it proposes a layer selection algorithm that minimizes a reformation of the error term as a surrogate loss. The proposed method has comparable model performance to full update, and saves computation and communication cost.

**Strengths:**

1. The theoretical part is very well-presented. The notations, definitions, and assumptions of this paper is very rigorous, and each lemma or theorem has good intuition.
2. The proposed method has stronger performance than baselines with fixed selection and baselines based on heuristics.

**Weaknesses:**

1. The exact training process is not very clear. In Algorithm 1, it seems each client only compute the gradient once, and the model is updated on the server, which is similar to FedSGD. However, In the analysis of computational cost, it seems like each client will first compute a step of full layer update, do the layer selection, and then do $$\tau$$ steps of selective layer update.
2. The computational cost seems over-simplified. In (16), the author simplify the cost of back-propagation as a simple sum of cost for each layer. However, this is generally not true. For example, for a ResNet network, finetuning the last linear layer is very efficient since one only needs to run back-propagation on the final layer. However, if one chooses to fine-tune the first convolutional layer, they need to back-propagate through the whole network to the very first layer, during which computing the gradient to intermediate feature maps. Therefore, just finetuning the last layer can save a lot of computation, while just finetuning the first layer may not save many computation. Also, the result in Table 3 may largely result from this, since in Figure 2(a), only the last few layers are selected after 10 epochs. Does the algorithm still save computation on other settings which may select the first layers (e.g., on XGLUE-NC)?
3. Lack of hyperparameter sensitivity w.r.t. $$\lambda$$, which seems to be related to gradient norms.
4. [Minor] Typos: Lemma 4.6 $$\nabla$$ in $$\mathcal{E}_{t,1}$$ should be $$\nabla_l$$​.
5. [Minor] Corollary 4.8 may not be very rigorous. Typically the learning rate is either a constant or a function of $$t$$, not a function of $$T$$. Also when $$\eta = \frac{1}{\sqrt{T}}$$, the third term seems limit to infinity. I believe it is better to just set learning rate to constant, and claim the third term dominate the RHS.

**Questions:**

1. How problem 1 (line 344) is optimized? It is an integer programming problem, which can require exponential complexity to get the globally optimal solution.
2. Problem 1 uses $$\sum_{i}\sum_{l} \| g_{i, l} \|_2^2$$ to approximate $$\mathcal{E}_{t, 1}$$. It looks weird to me since $$\sum_{l} \| \sum_{i} g_{i, l} \|_2^2$$ seems to be a closer approximation of the true $$\sum_l \| \nabla_l f \|_2^2$$. What is the beneficial of using the approximation in the paper?
3. See Weakness 1, what is the exact algorithm used?
4. Communication cost: since problem 1 requires pairwise comparison of client masks, it needs to be solved on server. It seems like in each communication round, each client needs to (1) compute the local gradient for all layers (2) upload all layers' gradients to the server, waiting for it to compute the mask m, (3) download mask m and perform selective layer updates, and (4) upload the gradients for partial layers. If it is true, the algorithm actually increase the communication cost. Please correct me if I am wrong.
5. For the CLIP model, which part of the model is updated? Only the vision encoder, or both vision and text encoder? Please provide these details in the Appendix if possible.

---

### Meta-Review · Area_Chair_3ygR · 2024-12-19

**Metareview:**

**Summary:** The paper investigates the impact of selective layer fine-tuning in federated learning settings. It provides a theoretical analysis of the relationship between selective layer updates and model convergence and proposes a gradient-based layer selection strategy to balance computational costs and heterogeneity across clients. Experiments conducted on both image and text datasets demonstrate the proposed approach's potential to save computation and improve adaptability compared to heuristic baselines.

**Decision:** This paper tackles an important problem in federated learning by exploring selective layer fine-tuning as a means to balance computation and heterogeneity. However, the proposed method lacks significant methodological innovation, as it builds on existing techniques without introducing distinct new ideas. Additionally, the theoretical contributions are generic and not directly tied to the proposed approach, weakening the overall impact. The experimental validation is limited to small-scale datasets and does not include comparisons to recent state-of-the-art methods. Besides, some reviewers also raise concerns about the further analysis of the results (e.g., sensitivity analysis of $\lambda$). These limitations outweigh the contributions, leading to the decision to reject. During the reviewer-AC discussion period, the reviewers unanimously agreed with this decision.

**Additional Comments On Reviewer Discussion:**

The authors did not respond to the reviewers' comments during the rebuttal period.

---

### Decision · Program_Chairs · 2025-01-22

Reject